# Heterogeneous Structural Disturbance of Cell Membrane by Peptides with Modulated Hydrophobic Properties

**DOI:** 10.3390/pharmaceutics14112471

**Published:** 2022-11-15

**Authors:** Yujiang Dou, Haibo Chen, Yuke Ge, Kai Yang, Bing Yuan

**Affiliations:** 1School of Electronic Information, Dongguan Polytechnic, Dongguan 523808, China; 2Center for Soft Condensed Matter Physics and Interdisciplinary Research, School of Physical Science and Technology, Soochow University, Suzhou 215006, China; 3Songshan Lake Materials Laboratory, Dongguan 523808, China

**Keywords:** peptide–membrane interaction, structural disturbance, AFM characterization, GUV leakage assay, molecular dynamics simulation

## Abstract

Extensive effort has been devoted to developing new clinical therapies based on membrane-active peptides (MAPs). Previous models on the membrane action mechanisms of these peptides mostly focused on the MAP–membrane interactions in a local region, while the influence of the spatial heterogeneity of the MAP distribution on the membrane was much ignored. Herein, three types of natural peptide variants, AS4-1, AS4-5, and AS4-9, with similar amphiphilic α-helical structures but distinct hydrophobic degrees (AS4-1 < AS4-5 < AS4-9) and net charges (+9 vs. +7 vs. +5), were used to interact with a mixed phosphatidylcholine (PC) and phosphatidylglycerol (PG) membrane. A combination of giant unilamellar vesicle (GUV) leakage assays, atomic force microscopy (AFM) characterizations, and molecular dynamics (MD) simulations demonstrated the coexistence of multiple action mechanisms of peptides on a membrane, probably due to the spatially heterogeneous distribution of peptides on the membrane surface. Specifically, the most hydrophobic peptide (i.e., AS4-9) had the strongest membrane binding, perturbation, and permeabilization effects, leading to the formation of large peptide–lipid aggregates (10 ± 5 nm in height and 150 ± 50 nm in size), as well as continuous fragments and ridges on the supported membrane surface. The AS4-5 peptides, with a half-hydrophilic and half-hydrophobic structure, induced membrane lysis in addition to reconstruction. The most hydrophilic peptide AS4-1 only exhibited unstable binding on the supported membrane surface. These results demonstrate the heterogeneous structural disturbance of model cell membranes by amphiphilic α-helical peptides, which could be significantly strengthened by increasing the degree of hydrophobicity and/or local number density of peptides. This work provides support for the modulation of the membrane activity of MAPs by adjusting their hydrophobicity and local concentration.

## 1. Introduction

Membrane-active peptides (MAPs) have attracted extensive interest in both fundamental biological research and practical biomedical applications [1,2,3,4]. Up to now, more than 2600 types of natural membrane-active antibacterial peptides with defined sequences and activities have been included in the antimicrobial peptide database (APD3) [5]. These peptides exhibit great potential for antibacterial, antiviral, anti-HIV, antifungal, antiparasitic, and anticancer uses [5]. In comparison with small-molecule drugs, MAPs show distinct advantages in inhibiting cellular targets, especially intracellular ones (e.g., inducing mitochondrial outer membrane permeability, MOMP) [6,7]. MAPs play a role by directly acting on target issues and/or affecting the host’s immune system. The unique abilities of MAPs have generated much interest in harnessing the properties of these peptides for the design of targeted medicine and/or to develop new therapies for wound healing, infectious diseases, and chronic inflammatory disorders [5]. To date, over 30 MAP-related drugs have been formulated for topical, oral, or intravenous uses in clinical trials [8,9].

Compared with the great progress in improving the functions of MAPs and the emerging potential clinical applications of MAP-based therapies, research on the interaction mechanism between MAPs and membranes is lagging [4]. Previous studies have shown that the action process of MAPs on a cell membrane has two stages, including the initial binding of peptides to the membrane surface, followed by the permeabilization of the membrane through insertion, poration, and/or disruption [10,11,12,13]. A variety of different mechanism models have been developed, such as the barrel–stave model, toroidal model, carpet model, and disordered toroidal model [14,15]. Different types of MAPs may adopt different membrane action modes. For example, caerin can achieve vertical transmembrane insertion using the barrel model [15] and melittin can form toroidal pores [16,17], and surfactants, such as polyoxyethylene (35) lauryl ether (Brij35), destroy membranes by lysis following the disordered toroidal model [18]. The development of advanced biophysical technologies in the last ten years provides deeper insight into the dynamic process of the interaction between peptides and a cell membrane, even at the molecular level [19,20,21,22,23]. For example, through combining the in-plane and Z-direction single-molecule tracking techniques, molecular dynamics (MD) simulations, and fluorescence imaging methods, we recently demonstrated that there are distinct metastable states in the membrane-insertion process of many MAPs (e.g., melittin, LL-37, PGLa, and Magainin) [20,24,25,26,27]. In particular, the membrane-insertion process of melittin is accompanied by conformational changes in the peptide, while PGLa and Magainin play distinct roles during their synergistic membrane poration process. Moreover, the toroidal pores formed by melittin are in dynamic equilibrium, with peptides continuously inserting into and extracting from the bilayer and lipids continuing to flip-flop between leaflets at the pore regions [20]. However, all of these models and studies describe the interaction process of MAP molecules and a membrane in a local region. For the whole large area of a cell membrane, the heterogeneity of MAP distribution in different regions may lead to the coexistence of some different action mechanisms.

The electropositive and amphiphilic properties are considered as the main factors determining the membrane activity of MAPs [28,29]. For example, the positive charges of peptides facilitate their binding to bacterial membranes (which have more negative charges), rather than the mammalian cell membranes, while the amphiphilic structure magnifies the disturbance of peptide binding to lipid packing states in the bilayer, which reduces the energy barrier for peptides to insert into the membrane [30]. In one of our recent studies, based on a natural membrane-active antimicrobial peptide, As-CATH4, we designed a series of peptide variants with different hydrophobic and electric charge properties, which showed distinct antimicrobial abilities [31]. In this work, three typical types of variants, AS4-1, AS4-5, and AS4-9 (Figure 1), with potent and comparable antibacterial efficiency but different hydrophobic characteristics were used to interact with model membranes. By sequentially substituting the hydrophilic residues, e.g., lysine (K) and arginine (R), at the polar–nonpolar interface of the amphiphilic α-helical structure of peptides with the more hydrophobic leucine (L) residue, the peptides demonstrate distinct degrees of hydrophobicity (AS4-1 < AS4-5 < AS4-9). By combining atomic force microscopy (AFM), dynamic giant unilamellar vesicle (GUV) leakage assay, and MD simulation methods, we found that the effect of peptide actions on a membrane may involve the coexistence of multiple mechanisms, including the spatially heterogeneous binding and aggregation of peptides on different regions of the membrane, and the consequent reconstruction and lysis of local areas of lipids. Such structural disturbance to membranes is strengthened by the enhanced degree of hydrophobicity (e.g., from AS4-1 to AS4-5 and to AS4-9), and/or increased local number density of peptides (e.g., with a peptide-to-lipid ratio from 1:50 to 1:14). This work deepens our understanding of the membrane action mechanism of amphiphilic α-helical MAPs and provides support for the design of MAP-based agents for biomedical uses.

## 2. Materials and Methods

### 2.1. Materials

1-palmitoyl-2-oleoyl-glycero-3-phosphocholine (POPC, referred to as PC here), 1-palmitoyl-2-oleoyl-sn-glycero-3-phospho-(1′-rac-glycerol) (POPG, referred to as PG here), and 1, 2-dipalmitoyl-sn-glycero-3-phosphoethanolamine-N-(lissaminerhodamine B sulfonyl) (RhB-PE) were purchased from Avanti Polar Lipids and used as received. Calcein was obtained from Sigma–Aldrich. Peptides with pre-designed amino acid sequences were ordered from SynPeptide Co., Ltd. (Nanjing, China; >95% purity). All experiments were carried out at room temperature at 22 °C.

### 2.2. Dynamic Giant Unilamellar Vesicle (GUV) Leakage Assay

GUVs were prepared following the conventional electro-formation method [25,32]. Briefly, a solution of lipids (PC: PG = 7:3 by mol, containing 0.5 wt% RhB-PE for fluorescence labeling; 60 μL × 2.0 mg mL^−1^ in chloroform) was deposited onto two ITO-coated glass slides and dried under a vacuum overnight. The dry film was transferred to a homemade electro-formation chamber (with the two glass slides as electrodes) and rehydrated in 0.1 M sucrose buffer. Alternating voltages were applied (0.5 V × 20 min, 1.0 V × 20 min, and 1.5 V × 3 h). The obtained vesicles were washed three times via centrifugation (8000 rpm × 20 min). Well-dispersed GUVs with a size distribution of 8−30 μm were collected, redispersed in 0.1 M sucrose buffer containing 0.2 mg mL^−1^ calcein (with a final lipid concentration of ∼0.02 mg lipid mL^−1^), and transferred to a sample chamber for observation.

A home-made chamber cell with a cover-glass substrate was used for microscope observation in the experiments. For the immobilization of the vesicles, the glass substrate was pre-treated with the following procedures. First, the glass slide was washed completely with ethanol and water, boiled in a mixture of H_2_O_2_ and H_2_SO_4_ (3:7 by vol) for 1 h, washed again with a large amount of water, followed by drying with N_2_ flow. It was then dipped into APTES for 5 min and dried under N_2_ flow again. After that, the glass slide was kept at 120 °C for 30 min before being installed in the chamber for use. For the in situ microscopy observations, a volume of the GUV dispersion was transferred to the chamber cell and stabilized for about 5 min for particle immobilization.

After the immobilization of GUVs on the substrate, a certain amount of peptide solution (to a final concentration of 2.0 μg mL^−1^) was added for in situ monitoring. Optical observations were performed on an inverted confocal laser scanning microscope (LSM 710, Zeiss) equipped with a 63× oil objective. Signals from the RhB channel for lipids (EX 543 nm, EM 575–640 nm), calcein channel (EX 488 nm, EM 530/50 nm), and the overlaid channel were captured simultaneously. The data were acquired with a pixel dwell time of 1.58 µs, and it generally took 3.87 s to take one image. Meanwhile, the transmission channel illuminated with a halogen lamp was acquired. All images were captured under the same instrumental settings. During the dynamic entry process of calcein into the interior of a GUV due to peptide exposure, the fluorescence intensity of the GUV interior at each time point (i.e., mean value among pixels read out directly from the Zeiss LSM software) was normalized with that of the surrounding environment and plotted as a function of time. More than five parallel experiments were performed in each condition, and representative GUVs were shown in the main text.

A variety of different peptide concentrations were used in the experiments (e.g., 1.0, 2.0, 5.0, and 10.0 μg mL^−1^). The membrane action effects of peptides exhibited an obvious concentration dependence. For the best comparison among the three types of peptides, a concentration of 2.0 μg mL^−1^ was finally adopted.

### 2.3. Atomic Force Microscopy (AFM) Characterization

Small unilamellar vesicles (SUVs) consisting of POPC were first prepared following the traditional extrusion method [33]. Briefly, PC lipids were dissolved in chloroform and then dried completely in a N_2_ stream. After that, they were rehydrated in Tris buffer (at 3.5 mg mL^−1^), ultrasonicated for 1 h, and extruded 21 times through a polycarbonate membrane with pores of 100 nm in size (Avanti Polar Lipids). Dynamic light-scattering (DynaPro, Malvern) measurements revealed that the obtained SUVs had a mean size of ~110 ± 15 nm. The SUV dispersion was then used to incubate a freshly exposed mica surface for 3 h before being washed away carefully. The solution was refreshed with ultrapure water for the following peptide incubation under a low-ionic-strength and neutral-pH condition. Due to the hydrophilic surface of the mica, a high-quality SLB was supposed to form [34], which was further confirmed by the continuous and homogeneous distribution of fluorescence of the film under a confocal fluorescent microscope.

Peptide exposure at 2.0 µg mL^−1^ for 40 min was carried out on SLBs at room temperature and refreshed with ultrapure water. AFM images of the supported membranes were collected with an Asylum Research MFP-3D-SA atomic force microscope setup in a tapping mode in the liquid phase. More than ten regions were characterized for each SLB, and more than five parallel experiments were performed under each condition. Representative images are shown in the main text.

### 2.4. Molecular Dynamics (MD) Simulations

The molecular structures of AS4-9 (LGLFKKLLRLILKGFKL) and AS4-5 (LGLFKKLLRLIKKGFKK) were built with the RPBS web resource (https://doi.org/10.1093/nar/gki477), which were further coarse-grained using the martinize.py script with the MARTINI2.2 force field [35,36]. A bilayer membrane composed of 600 PC and PG lipids (PC/PG = 7/3) was constructed by using the script insane.py [37]. The standard MARTINI water model was used [35], and the systems were neutralized using NaCl to a 0.15 M concentration. Initially, the peptides, with a peptide-to-lipid ratio of P/L = 1/14, 1/20, or 1/50, were randomly placed on the extracellular side of membranes.

The simulations were performed by using the GROMACS 2021 software package with the coarse-grained MARTINI force field [35,38] The isothermal–isobaric (NPT) ensemble at a temperature of 298 K and pressure of 1.0 bar was applied for all simulations. The temperature and pressure were maintained with the V-rescale scheme with a time constant of 1 ps and a Parrinello–Rahman semi-isotropic barostat with a time constant of 12 ps and compressibility of 3 × 10^−4^ bar^−1^. The periodic boundary conditions were applied in all three directions. All simulations were carried out for at least 2 μs with a time step of 20 fs after equilibrium. At least three independent runs were performed to ensure computational consistency. The order parameter of lipid chains was calculated following $\langle 3cos^{2}\theta -1\rangle /2$, and free energy was calculated by using metadynamics simulations [39].

## 3. Results and Discussion

### 3.1. Structure and Property of AS4-1, AS4-5, and AS4-9 Peptides

As-CATH4 was extracted from the Chinese alligator *Alligator sinensis* and first identified in 2017 [40]. It is composed of 38 amino acid residues, mainly adopts amphipathic α-helical secondary structures, and possesses potent antimicrobial activity against a variety of different strains. It has been suggested that the N-terminal 22 residues of As-CATH4 adopt α-helical conformation, and 17 of them (i.e., AS4) have been selected as a template to design peptide derivatives with different hydrophobicities and charging properties (Appendix A) [31]. According to previous studies [41], the positively charged lysine (K) and arginine^®^ residues at the polar–nonpolar interface of the amphiphilic α-helical structure were sequentially substituted by the more hydrophobic leucine (L) residue, based on which a series of peptides with varying hydrophobicities was obtained. The AS4-5 peptide, with a half-hydrophilic and half-hydrophobic structure, was first designed (Figure 1 and Appendix A). Correspondingly, the less hydrophobic peptide, AS4-1, and the more hydrophobic peptide, AS4-9, were adopted. The AS4-1, AS4-5, and AS4-9 molecules have net charges of +9, +7, and +5, respectively, while their hydrophobicity follows a reverse order of AS4-1 < AS4-5 < AS4-9. We have proven that all of these peptides have an α-helical secondary structure in a hydrophobic membrane mimetic environment (60 mM sodium dodecyl sulfate, as normally performed previously) [31,42]. Moreover, they exhibit potent and comparable antimicrobial activities, with minimal inhibitory concentration (MIC) values of 2.34, 2.34, and 1.17 μg mL^−1^ against *Staphylococcus aureus* ATCC25923 and MIC values of 4.69, 9.38, and 9.38 μg mL^−1^ against *Escherichia coli* ATCC25922, respectively [31]. The characteristically positively charged and amphiphilic properties of these peptides, with potent antibacterial ability, make them perfect candidates to study peptide–membrane interactions.

### 3.2. Membrane Permeabilization Ability of AS4-1, AS4-5, and AS4-9

The GUV leakage assay is normally used to test the membrane permeabilization ability of agents [25,32]. Herein, GUVs composed of mixed phosphatidylcholine (PC) and phosphatidylglycerol (PG) lipids. as generally used in previous studies [43,44] (at PC/PG = 7/3 by mol; labeled with 0.5 wt% rhodamine B-labeled phosphoethanolamine, RhB-PE), were dispersed in calcein solution. The GUVs remained stable without calcein entry for more than four hours under confocal observation (Appendix A). However, the addition of peptides (e.g., AS4-1, 5, or 9) induced obvious transmembrane entry of calcein from the outside to the inside of the GUVs. Figure 2 shows time evolution confocal images of representative GUVs after the addition of peptides (i.e., AS4-1, 5, or 9) at a fixed peptide concentration of 2.0 μg mL^−1^ (for details, refer to the Materials and Methods). The fluorescence intensity of the interior of the GUVs increased with time; meanwhile, the membranes kept in contact. This suggests the membrane permeabilization ability of the peptides. The transmembrane entry of calcein reached saturation within ~6 min after the addition of AS4-9; in contrast, this time was prolonged to ~35 min and ~60 min in the AS4-5 and AS4-1 conditions, respectively. This indicates a much higher membrane permeabilization efficiency of AS4-9 in comparison with AS4-5 and especially AS4-1.

For a more quantitative comparison, corresponding scatter diagrams referring to the time-dependent increase in fluorescence intensity in the interior of the GUVs were drawn (Figure 2D). It is interesting to note that the $\frac{I_{t}}{I_{\infty }}\sim t$ profile ($I_{t}$ and $I_{\infty }$ refer to the fluorescence intensities of the interior and surrounding environments of the GUV at *t* time, respectively) of AS4-9 can be fitted with a linear equation of $\frac{I_{t}}{I_{\infty }}=a+kt$ (*a* and *k* are constants), while that of AS4-5 or AS4-1 can be roughly fitted with a sigmoidal equation of $\frac{I_{t}}{I_{\infty }}=\frac{a}{1+\exp\left(-k\times \left(t-t_{c}\right)\right)}$ (*a*, *k*, and *t_c_* are constants) [32,45]. Herein, the linear dependence of $\frac{I_{t}}{I_{\infty }}$ on time indicates a constant entry rate of calcein into GUVs (i.e., constant *k*), suggesting that transmembrane pores or defects rapidly form on the GUV membrane within seconds upon AS4-9 addition. In contrast, the sigmoidal profile can be divided into three stages, including an initial negative exponent relation, followed by an approximately linear increase until reaching a saturation platform. This suggests an initially gradual accumulation and/or permeabilization process of peptides on membranes before the steady transmembrane diffusion of calcein through the formed defects or pores. The mean entry rate of calcein, calculated based on the linear state of individual $\frac{I_{t}}{I_{\infty }}\sim t$ profiles (marked with dashed lines in Figure 2D), confirmed the enhanced transmembrane leakage of calcein from AS4-1 to AS4-9 (Figure 2E).

Moreover, only under the AS4-9 condition, a bright green ring of calcein was observed, being well-colocalized with lipids after peptide actions; meanwhile, the deformation of some membrane regions was observed (marked with orange arrows in Figure 2A). These phenomena were generally observed in repeated experiments. This suggests that structural defects in lipid packing states have been produced by AS4-9. Considering the structural differences between AS4-9, 5, and 1, it is suggested that a more hydrophobic structure of peptides (although with less positive charges) could accelerate the membrane permeabilization process of peptides and produce stronger structural defects in lipid packing.

### 3.3. Structural Disturbance of Membranes by Peptides

To further investigate the peptide-induced structural disturbance to membranes, supported lipid bilayers (SLBs) were fabricated on the mica surface and exposed to peptides, after which the morphology of the membranes was characterized using AFM. Figure 3A–C shows representative images of a PC SLB after being incubated with AS4-9 at 2.0 μg mL^−1^ for 40 min. It is interesting to find that, in comparison with the smooth surface of a pristine SLB (Appendix A), the peptide-treated membrane exhibited different morphologies at different regions. Most of the surface (over 85%) exhibited small, dispersed particles with a height of 10 ± 5 nm (relative to the surrounding SLB surface) and a size of 150 ± 50 nm (Figure 3A). In some local areas (less than 10%), continuous fragments with a height of 5–7 nm and a size of a few micrometers were observed (Figure 3B). Occasionally, continuous ridges with a height of less than 6 nm and a length of a few micrometers were distinguished (Figure 3C). Such coexistence of different morphologies was generally observed in most specimens (Figure 3D–F), but was mostly ignored in previous reports [17,46]. Based on these observations, we can propose that the binding and aggregation of peptides on the membrane surface might induce the reconstruction of lipids in the membrane, leading to the formation of peptide–lipid complexes, demonstrated as the nanometer-sized particles dispersed on the SLB surface (Figure 3G). In some local areas, peptides adsorbed onto the membrane surface continuously and interacted with the lipids together, leading to the formation of a continuous fragment (similar to a carpet; Figure 3H). Occasionally, the adsorbed peptides were connected in a line, resulting in a linear bulge of the membrane (Figure 3I). The coexistence of different action mechanisms on a membrane may be caused by the heterogeneous distribution of the peptide number density in different regions of the membrane.

In comparison with AS4-9, the other two peptides (at the same peptide concentration of 2.0 μg mL^−1^) induced different structural disturbances to the membrane, which also deviated in different regions. Figure 4A–C shows representative AFM images of a SLB after being treated with AS4-5. Most of the SLB surface (over 50%) exhibited dispersed small particles with a height of 1~5 nm and a size of 130 ± 30 nm (Figure 4A). Parts of the membrane exhibited holes or ditches, with a depth of 4–5 nm (approaching the thickness of a bilayer membrane) relative to the surrounding SLB surface (Figure 4B,C). Moreover, particles with a height of ~5 nm were observed at the edge of some holes (marked with blue arrows in Figure 4B). These observations suggest that AS4-5 peptides can also adsorb and accumulate on the membrane surface; moreover, in some local areas, membrane lysis occurs due to peptide actions, leading to the formation of holes or ditches, with rough edges probably consisting of mixed peptides and lipids.

The AS4-1 peptides exhibited much weaker adsorption on the SLB surface compared with AS4-9 or AS4-5. As shown in Figure 4D–F, the SLB mainly displayed a smooth surface (being similar to a pristine membrane; Appendix A) with a small number of particles and short ridges (with a height of 4–5 nm) dispersed on the surface. Briefly, the AFM observations suggested that the disturbance of the membrane by peptides presented spatial heterogeneity and different mechanisms may coexist in different membrane regions. The more hydrophobic peptides (e.g., AS4-9) had stronger membrane adsorption, membrane disturbance, and membrane reconstruction capabilities, while the more hydrophilic peptides (e.g., AS4-5) had weaker membrane adsorption effects, but may have led to the lysis of local areas of the membrane. Too hydrophilic peptides (e.g., AS4-1) might only have undergone weak or unstable binding to the membrane surface.

### 3.4. Peptide-Induced Membrane Disturbance in MD Simulations

To further investigate the peptide-induced structural disturbance to membranes, MD simulations were performed. Different peptide-to-lipid ratios (e.g., P/L = 1/14, 1/20, or 1/50) were applied in the simulations to mimic the distinct peptide number densities at different membrane regions in the experiment. Initially, AS4-9 peptides were randomly placed on the extracellular side of PC/PG (7/3 by mol) membranes. It is interesting to note that the peptide–membrane interaction states were strongly dependent on the P/L ratios. Figure 5 shows the representative snapshots captured at the end of 2 μs simulations. With a higher P/L ratio of 1/14, obvious aggregation of the peptides was observed (marked with a dashed orange circle in the top view image in Figure 5A). Specifically, some lipids (shown in yellow and marked with a solid orange circle in the side view image) were further extracted from the bilayer by the aggregated peptides, suggesting the formation of peptide–lipid complexes. Lipid extraction was observed in the melittin–PC interaction system previously [25]. In contrast, with a lower P/L ratio of 1/20, linear distribution of the peptides was observed on the membrane surface (marked with dashed orange lines in Figure 5B). This heterogeneous distribution of peptides was also accompanied by a change in the membrane thickness, explaining the existence of particles, fragments, and ridges observed under AFM (Figure 3). In contrast, with an even lower P/L ratio of 1/50, only a dispersed distribution of peptides was observed (Figure 5C).

The interactions between AS4-5 and PC/PG membranes were also examined. Although only two residues of the sequence of AS4-5 were mutated, its membrane interaction states were obviously changed. For example, at the same P/L ratio of 1/14 (or 1/20), some of the AS4-5 peptides left the membrane surface and dispersed in the solution as aggregates (Figure 5D,E), rather than forming large peptide–lipid complexes in the membrane (Figure 5A). Calculations on the order parameter of lipid chains, following $\langle 3cos^{2}\theta -1\rangle /2$ [39], confirmed that AS4-5 produced weaker perturbation of the membrane than AS4-9 (Figure 5G). Enhanced interaction energy, both between peptides and lipids (Figure 5H), and between peptides (Figure 5I), was observed for AS4-9 compared with AS4-5, especially with high P/L ratios (marked with blue arrows), probably due to the higher degree of hydrophobicity of AS4-9 and the resultant, more complicated membrane interaction state (e.g., formation of peptide–lipid complex). Briefly, MD simulations proved the local number density-dependent distribution and accumulation of peptides on a membrane; moreover, the increased hydrophobicity of peptides (e.g., from AS4-5 to AS4-9) enhanced the membrane disturbance and strengthened the heterogeneity of the membrane structure after peptide actions.

It is interesting to note that the hydrophobicity of MAPs plays a key role in determining the MAP–membrane interaction mechanism, such as the binding and insertion of peptides on/into the membrane, as well as the spatial distribution and aggregation of peptides on/in the membrane. The membrane interface-facilitated aggregation (including oligomerization) mostly occurred for peptides with a specific degree of hydrophobicity. For example, it was predictable that overhydrophilic peptides (e.g., AS4-1) tended to disperse in the solution without stable membrane binding. In contrast, it is promising to find that, for the overhydrophobic peptides (e.g., AS4-9), in addition to aggregating into large particles, there were also some peptides that tended to disperse in the hydrophobic core layer of the membrane, probably due to the strong interaction energy between the peptides and lipids (top-view panel in Figure 5A). Moreover, membranes with more complex lipid compositions and/or an asymmetric bilayer structure might further complicate the MAP–membrane interaction situation [17].

## 4. Conclusions

Herein, three types of natural peptide variants, AS4-1, AS4-5, and AS4-9, which had similar electropositive properties and amphiphilic α-helical structures but different degrees of hydrophobicity (AS4-1 < AS4-5 < AS4-9) and net charges (+9 vs. +7 vs. +5) were used as model peptides to examine the effect of peptide properties (e.g., hydrophobicity) on the peptide–membrane interactions. Dynamic GUV leakage assays demonstrated that small variations in peptide sequences, as well as the degree of hydrophobicity, could strongly change the membrane permeabilization efficiency of the peptides. For example, AS4-9 exhibited the strongest ability to permeate the membrane compared with AS4-5 and especially AS4-1, with clear disturbance in the lipid packing state and deformation of the bilayer membrane. Moreover, AFM characterizations of the peptide-treated membranes (at 2.0 μg mL^−1^) showed different morphologies in different regions of the SLB membrane, suggesting the coexistence of multiple action mechanisms of peptides on a membrane during peptide–membrane interactions. For example, after exposure to AS4-9, the SLB membrane exhibited large particles (~10 nm in height and 150 nm in size), continuous fragments (~6 nm in height and a few μm in size), and ridges (≤6 nm in height); in contrast, the AS4-5-treated membrane exhibited fewer and smaller particles (1–5 nm in height and 130 nm in size) and holes/ditches referring to the local lysis of the bilayer; specifically, only a small number of particles and ridges (≤ 5 nm in height) was found on the AS4-1-exposed SLB surface. These observations suggest the strongest membrane binding and disturbance ability of AS4-9 (which is most hydrophobic in structure) and a membrane lysis ability of AS4-5 in addition to structural disturbance. MD simulations built up the correlation between the local number density and membrane disturbance effect of peptides, as well as the varying membrane interaction behaviors of peptide variants (e.g., formation of complex, fragments, and ridges). MAPs, either as innate immune defense components or as therapeutic agents, have shown a variety of different functions and attracted extensive commercial and research interest. Thoroughly understanding the interaction mechanism between MAPs and membranes will help the design of MAPs with cell-membrane-specific and modulated actions for clinical uses [47]. In this work, the combination of GUV leakage assays, AFM characterizations, and coarse-grained MD simulations provided the possibility to not only monitor the membrane permeabilization function of amphiphilic α-helical peptides, but also demonstrate the local (and whole) membrane structure disturbance due to peptide actions and reveal the molecular interactions between peptides and lipids. Although the contribution of different residues (especially those at the polar–nonpolar interface of the amphiphilic peptide) and different lipid compositions are not explained, this work reveals the coexistence of multiple action mechanisms caused by the spatially heterogeneous distribution of peptides on a membrane, which has been much ignored previously. Moreover, this work provides a strategy to effectively modulate the peptide-induced membrane structural disturbance by adjusting the degree of hydrophobicity and/or local number density of the peptides.

## Figures and Tables

**Figure 1 pharmaceutics-14-02471-f001:**
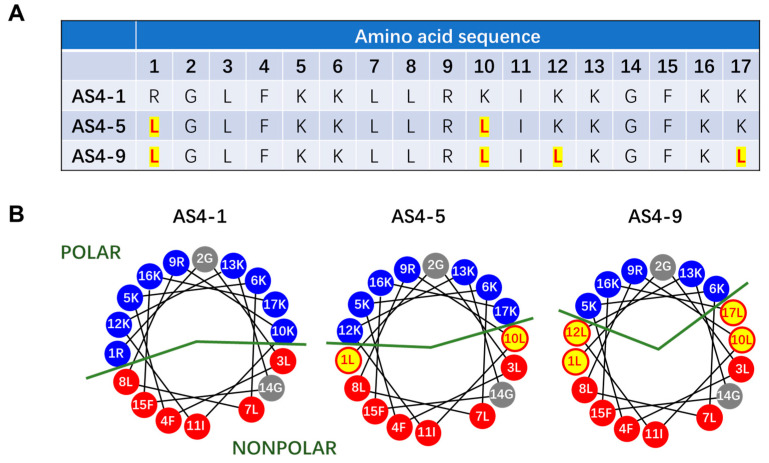
Amino acid sequences (**A**) and helical wheel representation (**B**) of the three types of peptides, i.e., AS4-1, AS4-5, and AS4-9. The mutated residues are emphasized in red with a yellow background. In (**B**), hydrophilic amino acids are shown with a blue background, and hydrophobic amino acids are shown with red and gray backgrounds. Green lines represent the interface between polar and nonpolar sides.

**Figure 2 pharmaceutics-14-02471-f002:**
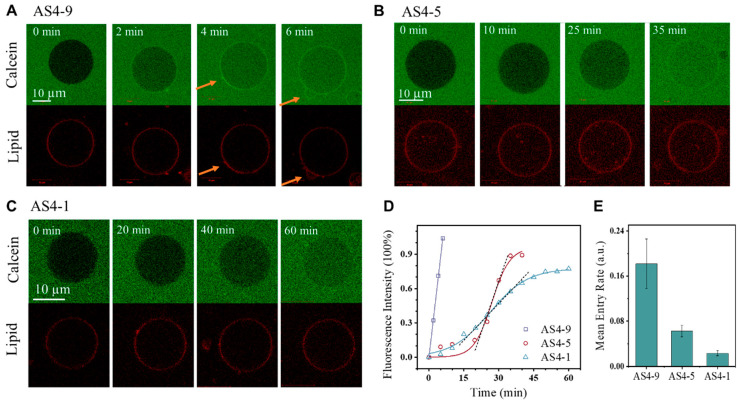
Peptide-induced membrane permeabilization. (**A**–**C**) Representative time series confocal images showing the transmembrane entry of calcein into a GUV due to peptide exposure. Images were taken in the green (calcein) and red (lipid) channels. (**D**) Corresponding scatter diagram showing the time-dependent increase in fluorescence intensity in the interior of the GUV due to drug-induced calcein entry. Fitting curves are shown with solid lines, following a linear equation of y=a+bx or a sigmoidal equation of $y=\frac{a}{1+\exp\left(-k\times \left(x-x_{c}\right)\right)}$. The linear stage of each profile (marked with dotted line) was used to calculate the mean entry rate of calcein (**E**). The membrane was composed of PC:PG = 7:3 by mol. The peptide concentration was 2.0 μg mL^−1^. Orange arrows in (**A**) refer to the deformed membrane. The data in (**E**) are based on at least three repeated tests.

**Figure 3 pharmaceutics-14-02471-f003:**
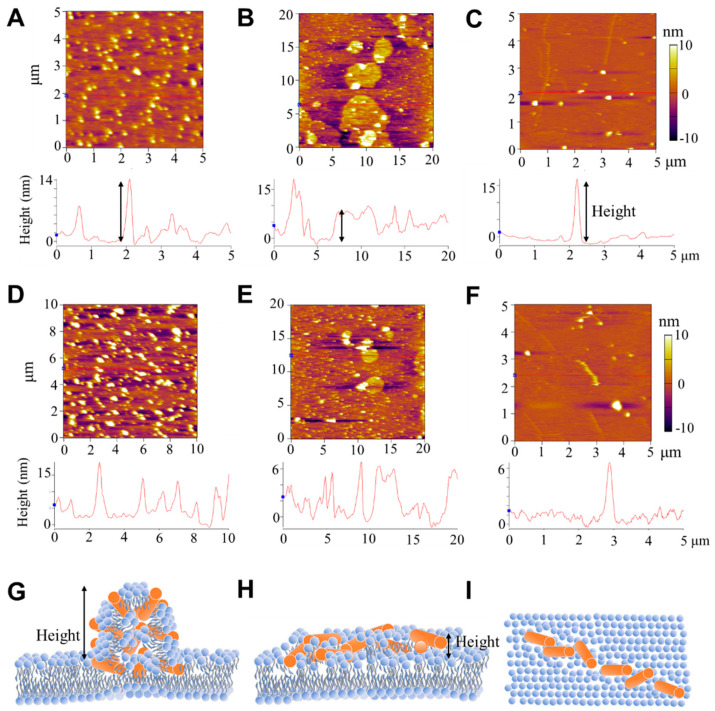
Representative AFM topographies of SLBs after AS4-9 exposure. Images (**A**–**C**) were obtained at different regions in a SLB membrane. (**D**–**F**) were captured from another sample. The corresponding height profiles are shown at the bottom. The orange background refers to the SLB surface, and the white regions refer to the aggregates/bulges on the surface. Before characterization, the membrane, consisting of PC lipids, was exposed to AS4-9 at 2.0 μg mL^−1^ for 40 min under a low-ionic-strength and neutral-pH condition. (**G**–**I**) Drawings showing the corresponding membrane structure in side (**G**,**H**) or top views (**I**). Orange rod: peptide; blue: lipid. Black arrows represent the height of the corresponding structures.

**Figure 4 pharmaceutics-14-02471-f004:**
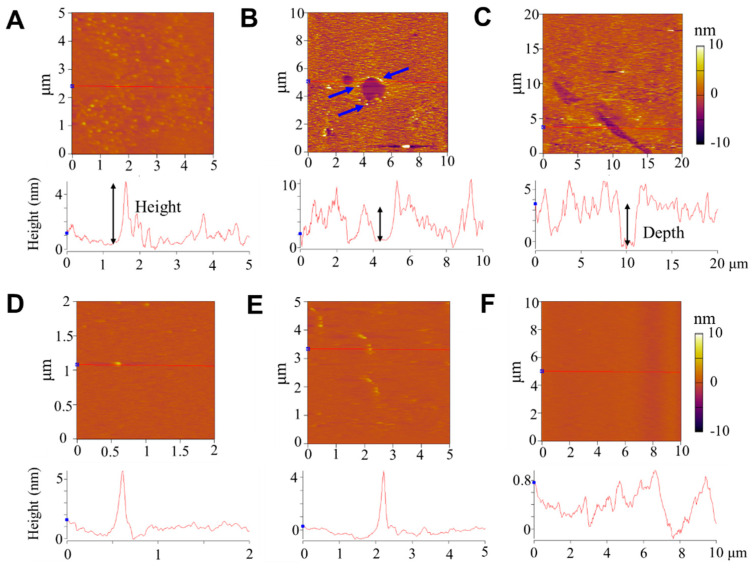
Representative AFM topographies of SLBs after AS4-5 (**A**–**C**) or AS4-1 (**D**–**F**) incubation. The images in (**A**–**F**) represent different membrane regions. The orange background refers to the SLB surface. Corresponding height profiles are shown at the bottom. The membrane, composed of PC, was incubated with peptides at 2.0 μg mL^−1^ for 40 min under a low-ionic-strength and neutral-pH condition before characterization. Blue arrows in (**B**) emphasize the raised edge of the hole. Black arrows represent the height or depth of corresponding structures.

**Figure 5 pharmaceutics-14-02471-f005:**
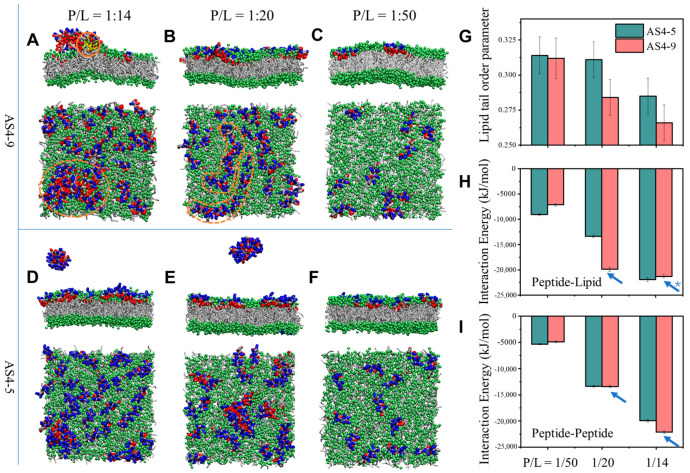
MD simulations of the interactions between peptides (i.e., AS4-5 and AS4-9) and a membrane (PC/PG = 7/3) at varying peptide/lipid (P/L) ratios. (**A**–**F**), representative snapshots. (**A**–**C**), AS4-9; (**D**–**F**), AS4-5. (**A**,**D**), P/L = 1/14; (**B**,**E**), P/L = 1/20; (**C**,**F**), P/L = 1/50. Green beads, lipid heads; silver lines, lipid tails. Blue and red colors refer to the hydrophilic and hydrophobic sides of the peptides, respectively. Dashed orange lines in (**A**,**B**) emphasize the aggregated or linear distribution of peptides. Solid orange circle in (**A**) refers to the extracted lipids (shown in yellow). (**G**–**I**) Histograms showing the order parameter of lipid chains (**G**) and the interaction energy between peptides and lipids (**H**) or between peptides (**I**) in different conditions. Blue arrows emphasize the stronger interaction energy for AS4-9 compared with AS4-5, and the asterisk indicates the influence of peptide aggregation. All the results were captured at 2 μs after peptide addition.

## Data Availability

The data presented in this study are available herein.

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
