# Peer review of "Heterogeneous Structural Disturbance of Cell Membrane by Peptides with Modulated Hydrophobic Properties"

_pharmaceutics, 2022, doi:10.3390/pharmaceutics14112471_

Round 1
Reviewer 1 Report
The manuscript contains interesting and valuable data on the interactions of MAP peptides (SA4-1,5,9) with the model PC/PG cell membrane system. In general, I have no fundamental comments on the methodology of the study, the results obtained, and the conclusions drawn from them. However, I noticed several defects in the text that should be corrected before publication.
1. The summary sentence ending the Abstract and Introduction sections is too general. I suggest replacing it with a more informative one.
2. in the Results section:
The philosophy underlying the design of AS4-5.9 peptides should be better described. This is not a "simple" sequential substitution of basic amino acid (aa) residues (R,K) with hydrophobic ones (L) at the boundary between hydrophilic and hydrophobic regions. The substitution limited to 1 or 2 residues on the N- and C-terminal side of this boundary. Why was the substitution limited to only a maximum of two residues on each side of this area? This is not explained by the authors.
The authors write that an earlier conformational analysis performed in SDS (ref. 31) showed an alpha helical conformation of the peptides studied. However, the environment studied in the manuscript was the PC/PG system, not SDS. The results in the two environments should not be qualitatively different (quantitatively they might). The authors do not clarify this issue in their paper by assuming that the peptides behave identically in both environments. Please address this issue when analyzing helical wheel projection of peptides.
Please indicate in the text of the manuscript that the AS4-1 sequence is the native sequence.
There is a duplication of the POPC-PC and POPG-PG lipid labeling in the text. Please opt for one variant.
3. in the Materials and methods section
Any description of peptide synthesis and characterization (HPLC, MS) is missing. These data can be found in Supporting Information Hydrophobicity determines the bacterial killing rate of α-helical antimicrobial peptides and influences the bacterial resistance development by Minghui Zhang et al. but this is not referenced in the manuscript.
Author Response
Please refer to the separate file.

Reviewer 2 Report
Dear authors,
The manuscript entitled “Heterogeneous structural disturbance of cell membrane by peptides with modulated hydrophobic properties” aimed to to develop new clinical therapies based on membraneactive peptides (MAPs). This study is an extension of preliminary work “Hydrophobicity Determines the Bacterial Killing Rate of α‑Helical Antimicrobial Peptides and Influences the Bacterial Resistance Development” published in the “Journal of Medicinal Chemistry”. It presents scientific relevance for the area of Pharmacy, Chemistry, Medicine and others area of the healthy. After consulting at www.sciencedirect.com; https://pubmed.ncbi.nlm.nih.gov/ and others databases, some authors have publications related to subjects related to the theme of the manuscript. The language (English) are satisfactory (I suggest the final revision)! However, you need to change some details/informations in the “abstract”, “Introduction”, “methods”, “results”, “discussion” and “conclusions”.
Abstract: Adequate, but I suggest:
- The abstract is well written. However, I suggest informing the main objective of the study, right after the “background”.
- I suggest highlighting the "innovative" proposal of the study at the end of the abstract.
- The keywords “peptide-membrane interaction” and “molecular dynamics simulation” are not included in the title or abstract. I suggest review!
1. Introduction section: It is well written, but:
- Figure 1 is quoted at the end of the introduction and Page 3 (results and discussion)! I suggest presenting Figure 1 (and others) immediately after its citation in the text!
- I suggest highlighting the "innovative" proposal of the study, as well as the advantages / disadvantages or limitation, at the end of the introduction.
2. Results and discussion section:
- Page 5, in “2.3. Structural disturbance of membranes by peptides” section: Are these data inedited/unpublished? I suggest comparing the results obtained with data from the literature! Idem for “2.4. Peptide-induced membrane disturbance in MD simulations” section!
- Page 8: After Figure 5, I suggest expanding further discussions on topic (2.4. Peptide-induced membrane disturbance in MD simulations). I suggest inserting a paragraph at the end of the subsection highlighting the main ideas discussed and perspectives.
3. Conclusions section:
- I suggest rewriting, improving the conclusions based on some comments. I suggest highlighting the advantages of the method and the study! At the end of the section, I suggest highlighting the importance of the study area and limitations!
4. Materials and Methods section: The proposed methods are suitable for the study:
- Page 9, in “4.2. Dynamic giant unilamellar vesicle (GUV) leakage assay”: How were the studied variables optimized? Were the methodological protocols created by the authors? Or did they follow previous studies already published? I suggest indicating some references to the indicated methods.
* Tables and Figures: Adequate!
* References: Please, check if the references are in accordance with the journal's rules.
Author Response
Please refer to the separate file.
